# Patch Matching and Dense CRF-Based Co-Refinement for Building Change Detection from Bi-Temporal Aerial Images

**DOI:** 10.3390/s19071557

**Published:** 2019-03-31

**Authors:** Jinqi Gong, Xiangyun Hu, Shiyan Pang, Kun Li

**Affiliations:** 1School of Remote Sensing and Information Engineering, Wuhan University, Wuhan 430079, China; jinqigong@whu.edu.cn (J.G.); petrick_Lee@whu.edu.cn (K.L.); 2Collaborative Innovation Center of Geospatial Technology, Wuhan University, Wuhan 430079, China; 3School of Educational Information Technology, Central China Normal University, Wuhan 430079, China; psy@whu.edu.cn

**Keywords:** building change detection, patch matching, phase consistency, semantic segmentation, relief displacement

## Abstract

The identification and monitoring of buildings from remotely sensed imagery are of considerable value for urbanization monitoring. Two outstanding issues in the detection of changes in buildings with composite structures and relief displacements are heterogeneous appearances and positional inconsistencies. In this paper, a novel patch-based matching approach is developed using densely connected conditional random field (CRF) optimization to detect building changes from bi-temporal aerial images. First, the bi-temporal aerial images are combined to obtain change information using an object-oriented technique, and then semantic segmentation based on a deep convolutional neural network is used to extract building areas. With the change information and extracted buildings, a graph-cuts-based segmentation algorithm is applied to generate the bi-temporal changed building proposals. Next, in the bi-temporal changed building proposals, corner and edge information are integrated for feature detection through a phase congruency (PC) model, and the structural feature descriptor, called the histogram of orientated PC, is used to perform patch-based roof matching. We determined the final change in buildings by gathering matched roof and bi-temporal changed building proposals using co-refinement based on CRF, which were further classified as “newly built,” “demolished”, or “changed”. Experiments were conducted with two typical datasets covering complex urban scenes with diverse building types. The results confirm the effectiveness and generality of the proposed algorithm, with more than 85% and 90% in overall accuracy and completeness, respectively.

## 1. Introduction

Automatic building change detection (BCD) from aerial images is a relevant research area in the remote sensing field, as the results are required for a range of applications such as urbanization monitoring, identification of illegal or unauthorized buildings, land use change detection, digital map updating, and route planning [1]. Information about changes in buildings can be useful for aiding municipalities with long-term residential area planning. BCD is used to analyze the condition of damaged buildings after earthquakes and other natural disasters, supporting rescue activities and reconstruction measures [2]. With the development of remote sensing techniques, an ever-growing number of remote sensing images need to be processed [3]. As manual processing heavily relies on human interpretation and is a labor-intensive and time-consuming work, unsupervised techniques are required to perform BCD without the availability of ground truth [4].

Many BCD approaches and techniques for handling remote sensing data have been developed, and newer techniques are still being created. Usually, BCD involves two main procedures: building change generation (BCG) and segmentation of the building change map. As the core part of BCD, BCG aims to highlight changes in the buildings, and it directly affects the accuracy. Segmentation is used to distinguish the changed from unchanged pixels by transforming the building change map into a binary map, which facilitates the evaluation of the accuracy of the BCD.

A variety of different algorithms have been proposed for the automatic detection of changes based on bi-temporal or multi-temporal remote sensing images. They vary from pixel-oriented methods to object-oriented methods, and from spectral-characteristics-based methods and artificial-intelligence-based methods [5]. Conventional pixel-oriented methods mainly use techniques based on algebraic operations [6], transformation [7], and classification [8] to recognize change information. These methods have been confirmed to detect abrupt changes from low- or medium-resolution images [9]. With the increasing availability of high-resolution remote sensing images, it is necessary to detect detailed changes occurring at the level of the ground structures, including buildings. Conventional pixel-based methods may cause many small pseudo changes because of the increased high-frequency components [10], and another limitation in automatic detection of changes is that modeling the contextual information is difficult. To address these problems, spatial dependence among neighboring pixels, e.g., object, textural- or structural-based image description, have been used in BCD [11]. A set of novel building change indices (BCIs) were proposed that combine the morphological building index (MBI) and slow feature analysis (SFA) for change detection from high-resolution imagery [12]. The proposed method does not need any training samples and can reduce human labor. Tang et al. [13] proposed using geometrical properties, including the interest points and structural features of buildings, to identify the building changes from multi-temporal images. The proposed method is insensitive to the geometrical differences of buildings caused by different imaging conditions in the multi-temporal high-resolution imagery and is able to significantly reduce false alarms.

Object-oriented methods, evolved from the concept of object-based image analysis [14], not only employ the spectral, texture, and transformed values, but also exploit extra information about the shape features and spatial relations of objects using image segmentation techniques. As image objects are used as the basic units in object-oriented methods, they are more suitable for handling high-resolution remotely sensed images and can achieve better performance [15,16,17,18,19]. Xiao et al. [17] presented a co-segmentation-based method for building change detection providing a new solution to object-based change detection. Their method takes full advantage of multi-temporal information and produces two spatially-corresponded change detection maps using the association of the change feature with image features, and the method can reveal the thematic, geometric, and numeric changes in the objects. A saliency-guided semi-supervised building change detection method was proposed in Hou et al. [20], which combines pixel-based post-classification with object-based semi-supervised classification and produced promising results on challenging datasets. Huang et al. [21,22] investigated urban building change in an object-based environment by integrating MBI, morphological shadow index (MSI), and spectral and shape conditions from multitemporal high-resolution images. An enhanced morphological building index was proposed and used for building change detection with a change vector analysis method in Feng et al. [23]. The index not only removes the noise in the homogeneous regions but also improves detection accuracy. Liu et al. [24] presented a line-constrained shape (LCS) feature that more easily distinguishes buildings from other geo-objects. Then, based on LCS and spectral features, an object-based supervised classification method was used for BCD.

Approaches have been proposed using deep neural networks to solve imagery interpretation and change detection problems, and have achieved good performance [25,26,27,28]. Zhang et al. [29] proposed and built a model based on Gaussian–Bernoulli deep Boltzmann machines with a label layer to learn high-level features, which was trained for determine the change areas. Gong et al. [3] designed coupled dictionary learning (CDL) to explore the intrinsic differences in multisource data for change detection in a high-dimension feature space, and proposed an iterative scheme for unsupervised sample selection to retain the purity of training samples and gradually optimize the current coupled dictionaries. Although the methods based on deep learning have advantages compared with traditional vision algorithms in semantic segmentation and object detection for remotely sensed images, they also have some deficiencies. First, vast amounts of training data are usually required for training and few annotated building change detection datasets are available that can be used to train supervised machine learning systems detecting changes in image pairs. Deep neural networks are complicated models, so powerful computational facilities are usually required for the training process. In addition, considerable effort is required to adjust some hyper-parameters for a good model, which requires time and computational power because the model performance depends on accurate parameters [30].

Despite some efforts to develop building change detection techniques, spectral variation, relief displacement, and the composite structure of buildings in the aerial images complicate obtaining highly accurate results with BCD. The surface of the earth is not smooth and flat. As a consequence, this natural phenomenon disrupts the true orthogonality of photo image feature. On an aerial photograph, the displacement of the image due to variation in the terrain relief is known as relief displacement or height distortion [31]. Figure 1a illustrates the relief displacement caused by differences in the relative elevation of objects photographed. If the higher object is viewed from infinity, its image is a point and no relief displacement exists. If it is viewed from a finite altitude, its image appears to “lay back” on the adjacent terrain, and the displacement vector in the image is ***aa’***. The magnitude of this displacement vector depends on the height of the object, the flying height, and its location in the image. Such displacements in the image are always radial from the image of the nadir point. The nadir point is the point exactly beneath the perspective center. Even if an image is geometrically corrected (orthorectified) with a digital elevation model (DEM), relief displacement cannot be completely removed as the height of ground objects is not considered. Especially for high-rise buildings in high-resolution remote sensing images, their rooftops largely deviate from the ground, appearing as oblique relief displacements, and huge differences among multi-temporal images are always observed, as shown in Figure 1b. The use of true-ortho rectification [32] may solve this displacement problem, but it requires a high-resolution digital surface model (DSM) and additional processing. Some erroneous results may be caused by spectral variation between bi-temporal images and an unexpected appearance of heterogeneity due to objects located on building roofs, such as pipes, occlusions, shanties, etc., leading to low quality and inaccurate change detection. 

Given this context, considering heterogeneous appearances and positional inconsistencies is necessary, particularly when detecting changes in buildings with composite structures and relief displacements. Therefore, we propose a novel method based on patch matching and densely connected CRF optimization for co-refinement to detect building changes from bi-temporal aerial images. In this approach, we first obtain the bi-temporal changed building estimates using a graph-cuts-based segmentation algorithm integrating the change information and extracted buildings. Then, a structural feature descriptor, histogram of orientated phase congruency (HOPC) including corner and edge information, is used to perform patch-based matching in the proposed bi-temporal changed buildings. Next, on the basis of co-refinement with CRF gathering matched roof and bi-temporal building features, we determined the change type of a building: “newly built,” “demolished”, or “changed”. Two typical datasets of bi-temporal aerial images were used to verify the effectiveness and generality of the proposed method.

The main contributions of this paper are: (1) in the generation of proposed bi-temporal building changes. The segmentation procedures are associated with the same change information; they take full advantage of the extracted building features at each phase in the deep learning based semantic segmentation. This is a state-of-the-art method producing reliable initial information of building locations and changes. (2) The corresponding roofs obtained by an effective patch-based matching approach represent the unchanged area, combining the information of potential changed buildings in the bi-temporal images, which spontaneously eliminates the matching of inconsistent unchanged buildings caused by the relief displacement and spectral variation. Additionally, (3) during the process of co-refinement based on CRF, our method can integrate the matching probability, color contrast, and spatial distance of bi-temporal images to better determine the changes.

This article is organized as follows: Section 2 presents the proposed method. The experimental assessment and a discussion of the obtained results are presented in Section 3 and Section 4, respectively. Finally, conclusions from an analysis of the obtained experimental results are presented in Section 5.

## 2. Methodology

A novel building change detection approach based on patch matching from bi-temporal aerial images is presented in this work, and the processing work flow is shown in Figure 2. To implement the entire framework, the proposed method consists of two parts: generation of proposed building changes (Section 2.1.) and patch-based roof matching (Section 2.2.), and co-refinement for final building change detection (Section 2.3). 

### 2.1. Generation of Changed Building Proposals 

To generate the changed building proposals, the change information is first obtained with the object-oriented method using iteratively reweighted multivariate alteration detection (IR-MAD) [33]. Then, convolutional neural network (CNN)-based semantic segmentation is adopted to extract buildings. Finally, segmentation is performed via graph-based energy minimization under the guidance of the change information, combined with the extracted buildings, directly resulting in bi-temporal changed building proposals.

#### 2.1.1. Object-Oriented Change Detection Using IR-MAD

For high-resolution remote sensing images, superpixels are considered to be the basic unit against the scatter of the change information and are the effect of the salt and pepper noise. Several commonly used algorithms are available for superpixel segmentation, such as simple linear iterative clustering named SLIC [34], MeanShift [35], watershed [36], etc. The superpixels determined by the SLIC algorithm have good consistency and compactness, adhere to boundaries, and are straightforward to extend to superpixel generation [34]. The images to be segmented in this paper involved different periods of data, and multi-temporal image-object is considered to be the most appropriate analysis unit for change detection. In multispectral change detection, IR-MAD is proving to be accurate, since various alternatives exist in which the weights (no-change probabilities) are calculated during the iteration procedure. Thus, we designed an object-oriented change detection technique to obtain the superpixel (i.e., bi-temporal image-object) with the SLIC algorithm and to generate change information using IR-MAD. 

We first segmented the bi-temporal images together and obtained bi-temporal superpixels (i.e., bi-temporal image-object) for comparison and analysis, as shown in Figure 3a–c. The bi-temporal image-object [37] considers all series of the data simultaneously; thus, it has the distinct advantage of maintaining a consistent topology and potentially maintaining key multi-temporal boundaries. Then, the region property descriptor, corresponding to the homogeneous image-object, is extracted from the bi-temporal aerial images to build two sets of vectors in feature space. The region property descriptor depicts an object by its color, texture, and structure information, and includes the means of red-green-blue (RGB), local binary pattern (LBP) [38], Gabor-filtered [39] values in four directions, and entropy and energy. Next, multivariate alteration detection (MAD) variates are calculated using the differences between canonical variates from canonical correlation analysis (CCA) on the two feature vectors. Finally, an iterative re-weighting scheme is proposed to further enhance the change information, and a probability density function of chi-square distribution is introduced to obtain invariant objects. This procedure produces an increasingly better background for no change, against which change can be measured. More details are provided in Canty et al. [40,41]. To improve the visibility and clarity of change information, reverse operation and normalization processing are applied to produce a change confidence index (CCI), as shown in Figure 3d.

#### 2.1.2. Building Extraction with Semantic Segmentation

Deep learning techniques have been widely used for image analysis for many years. Most notably, convolutional neural networks (CNNs) are a family of algorithms that are especially suited for working with images. Semantic image segmentation has recently witnessed considerable progress by training deep CNNs. As the dual multi-scale manifold ranking (DMSMR) network [42] estimates the predicted labels in an end-to-end fashion, and the dilated and non-dilated convolution layers are jointly optimized by manifold ranking. The network combines dilated, multi-scale strategies with the single stream manifold ranking optimization method in the deep learning architecture to further improve the performance without any additional aides. Thus, we adopted the DMSMR network-based semantic segmentation to label each pixel and segregate them. In this work, the EvLab-SS dataset from our team, which is designed for the high-resolution pixel-wise classification task on real engineered scenes in remote sensing areas, is used to train the deep CNN for DMSMR. The dataset is originally obtained from the Chinese Geographic Condition Survey and Mapping Project, and each image is fully annotated by the Geographic Conditions Survey standards. The average resolution of the dataset is approximately 4500 × 4500 pixels. The EvLab-SS dataset contains 11 major classes, namely, background, farmland, garden, woodland, grassland, building, road, structures, digging pile, desert and waters, and currently includes 60 frames of images captured by different platforms and sensors. We produce the training dataset by applying the sliding window with a stride of 128 pixels to the training images, resulting in 48,622 patches with a resolution of 640 × 480 pixels. Similar methods are utilized on validation images, thus generating 13,539 patches for validation. Figure 4 gives some examples of the training data for building extraction. In terms of model parameter settings, they are completely consistent with those in [34], and more details can be seen in Zhang et al. [34]. Other CNN-based semantic segmentation algorithms for aerial images are also suitable for building detection. Semantic segmentation using the DMSMR network for bi-temporal images is shown in Figure 5a,b. As this paper focuses on building change detection, only the category of buildings is labeled 1, and all the other categories are assigned 0. The result of building extraction with sematic segmentation is called sematic building label (SBL), and SBLs on bi-temporal aerial images are shown in Figure 5c,d, respectively.

#### 2.1.3. Segmentation Based on Graph Cuts

Given the work above, both the change information and the extracted buildings at each phase were combined in a graph-based energy function, and we used the graph cuts [43] that adopt a max-flow/min-cut algorithm [44] to find the optimal solution by constructing a weight map to perform segmentation for the generation of bi-temporal changed building proposals. The energy function of a segmentation based on graph cuts is defined as:(1)$$E=E_{data}+E_{smooth}$$
where $E_{data}$ represents the data term that helps find the potential change region, which is expressed as:(2)$$E_{data}=\underset{p\in P}{\sum }D_{p}\left(l_{p}\right)$$
(3)$$D_{p}\left(l_{p}\right)=\{\begin{array}{c} -\ln C_{p},\;if\;l_{p}=1\\ -\ln\left(1-C_{p}\right),\;otherwise \end{array}$$
where $D_{p}\left(l_{p}\right)$ represents the cost for assigning pixel p to a label that is either foreground ($l_{p}=1$) or background ($l_{p}=0$), P is a set of all the pixels (i.e., nodes in the graph), and $C_{p}$ indicates the change probability of pixel p being labeled as building, which is obtained by the operation CCI×SBL. Hence, the change information not only provides the prior knowledge of building changes, in which a larger value implies a more probable change, but serves as an association for separate segmentation procedures to generate the bi-temporal changed building proposals. 

$E_{smooth}$ denotes the smooth term, which is mainly used to penalize the discontinuity between neighborhood pixels. A segmentation boundary occurs when two neighboring pixels are assigned different labels. Most nearby pixels are expected to have the same label; therefore, no penalty is assigned if neighboring pixels have the same label, and a penalty is assigned otherwise. Usually, this penalty depends on RGB difference between pixels, which is small in regions of high contrast [45]. Thus, we define the smooth term as used in Rother et al. [46]:(4)$$E_{smooth}=\underset{\left\{p,q\right\}\in N}{\sum }V_{\left\{p,q\right\}}\left(l_{p},l_{q}\right)$$
(5)$$V_{\left\{p,q\right\}}\left(l_{p},l_{q}\right)=\{\begin{array}{c} \max\left(\lambda ,1\right),\;if\;S_{p}=1\;and\;S_{q}=1\\ \exp\left(-\frac{{||I_{p}-I_{q}||}^{2}}{2\sigma^{2}}\right)\times \frac{1}{d\left(p,q\right)}\times \lambda ,\;otherwise \end{array}$$
where N is the set of all pixel pairs in the neighborhood (i.e., edges in the graph), p and q are two neighboring points, $S_{k}$ denotes the semantic label of the pixel in the SBL, $I_{k}$ represents the pixel color, d(p,q) is calculated as the Euclidean distance between pixels p and q, and $V_{\left\{p,q\right\}}\left(l_{p},l_{q}\right)$ defines the cost of assigning the labels $l_{p}$ and $l_{q}$ to the pixel pairs p and q, respectively. The weight coefficient λ>0 specifies the relative importance between the data term and the smooth term, and $\sigma^{2}$ is a scale parameter, which is set as suggested by Rother et al. [46]:(6)$$\sigma^{2}=\langle {||I_{p}-I_{q}||}^{2}\rangle$$
where 〈·〉 is the average value over the whole image.

According to the designed energy function, the max-flow/min-cut algorithm is performed to segment the bi-temporal images into foreground and background, respectively. Figure 6 depicts the changed building proposals (CBP) in the bi-temporal aerial images.

### 2.2. Patch-Based Matching

After generating the bi-temporal changed building proposals, many building changes were mislabeled, which was caused by heterogeneous appearances and different relief displacements between the bi-temporal aerial images. Therefore, we needed to further determine the corresponding relationship of the roofs between the bi-temporal aerial images to impose a constraint on the building change detection. The basic aim of patch-based roof matching is to estimate the corresponding relation of the roofs between bi-temporal aerial images, which includes collecting corner and edge information for feature detection through the PC model and feature matching with the structural HOPC descriptor. As the buildings are quite similar on a local scale in terms of height, contour shape, and structure, we used robust filtering strategies for estimating geometric transforms and improving the stability of the process.

#### 2.2.1. Feature Detection 

Classical feature detectors generally rely on image intensity or gradient information, which is spatial domain information, such as Sobel, Canny, and scale-invariant feature transform (SIFT). In addition, image features can be described using frequency domain information, such as phase information. By comparison, phase information is more robust to image illumination, scale, contrast, and other changes. Morrone et al. [47] observed that highly consistent local phase information is usually present in certain points of the image, causing a strong visual response. Thus, phase congruency (PC) is important for image perception and using it for feature detection is logical. 

Gabor filters are a traditional choice for obtaining localized frequency information. They offer the best simultaneous localization of spatial and frequency information. However, they have two main limitations. The maximum bandwidth of a Gabor filter is limited to approximately one octave and Gabor filters are not optimal if seeking broad spectral information with maximal spatial localization. An alternative to the Gabor function is the log-Gabor function proposed by Field [48]. Log-Gabor filters can be constructed with arbitrary bandwidth and the bandwidth can be optimized to produce a filter with minimal spatial extent. On the linear frequency scale, the log-Gabor function has a transfer function in the form:(7)$$g\left(w\right)=\exp^{\left(-\log{\left(\omega /\omega_{0}\right)}^{2}\right)/\left(2(\log{\left(\sigma_{\omega }/\omega_{0}\right)}^{2}\right)}$$
where $\omega_{0}$ is the filter’s central frequency and $\sigma_{\omega }$ is the related width parameter. The corresponding spatial domain filter of the log-Gabor wavelet can be obtained using inverse Fourier transform. The real and imaginary parts of the filter are referred to as the log-Gabor even-symmetric and odd-symmetric wavelets, respectively. Thus, given an input image *I(x, y)*, the response components can be calculated by convolving *I(x, y)* with the two wavelets:(8)$$\left[E_{so}\left(x,y\right),O_{so}\left(x,y\right)\right]=\left[I\left(x,y\right)*L_{so}^{even},I\left(x,y\right)*L_{so}^{odd}\right]$$
where $E_{so}\left(x,y\right)$ and $O_{so}\left(x,y\right)$ are the responses of the log-Gabor even-symmetric $L_{so}^{even}$ and odd-symmetric $L_{so}^{odd}$ wavelets at scale s and orientation o, respectively. Then, the amplitude $A_{so}$ and phase $\phi_{so}$ at scale s and orientation o are obtained by:(9)$$A_{so}\left(x,y\right)=\sqrt{E_{so}{\left(x,y\right)}^{2}+O_{so}{\left(x,y\right)}^{2}}$$
(10)$$\phi_{so}\left(x,y\right)=\arctan\left(O_{so}\left(x,y\right)/E_{so}\left(x,y\right)\right)$$

Considering the noise compensation, the final PC model with all scales and orientations in Kovesi et al. [49] is defined as: (11)$$PC\left(x,y\right)=\frac{\sum_{s}\sum_{o}W_{o}\left(x,y\right)⎣A_{so}\left(x,y\right)\Delta \Phi_{so}\left(x,y\right)-T⎦}{\sum_{s}\sum_{o}A_{so}\left(x,y\right)+\varepsilon }$$
where $W_{o}\left(x,y\right)$ is a weighting function, ε is a small constant, and the ⎣ ⎦ operator denotes that the enclosed quantity is non-negative, meaning the enclosed quantity is equal to zero when its value is negative. $\Delta \Phi_{so}\left(x,y\right)$ is a phase deviation function, whose definition is:(12)$$A_{so}\left(x,y\right)\Delta \Phi_{so}\left(x,y\right)=\left(E_{so}\left(x,y\right){\bar{\phi }}_{E}\left(x,y\right)+O_{so}\left(x,y\right){\bar{\phi }}_{O}\left(x,y\right)\right)\;\;-\left|E_{so}\left(x,y\right){\bar{\phi }}_{O}\left(x,y\right)-O_{so}\left(x,y\right){\bar{\phi }}_{E}\left(x,y\right)\right|$$
(13)$${\bar{\phi }}_{E}\left(x,y\right)=\sum_{s}\sum_{o}E_{so}\left(x,y\right)/C\left(x,y\right)$$
(14)$${\bar{\phi }}_{O}\left(x,y\right)=\sum_{s}\sum_{o}O_{so}\left(x,y\right)/C\left(x,y\right)$$
(15)$$C\left(x,y\right)=\sqrt{{\left(\sum_{s}\sum_{o}O_{so}\left(x,y\right)\right)}^{2}+{\left(\sum_{s}\sum_{o}E_{so}\left(x,y\right)\right)}^{2}}$$

Although this model produces a PC measure that results in a very good edge map, it ignores information about the way in which PC varies with orientation at each point in the image. To address this problem, we first produced an independent PC map $PC\left(\theta_{o}\right)$ for each orientation o using Equation (15), and the following three intermediate quantities are calculated according to the classical moment analysis equations [50]:(16)$$a=\sum_{o}{\left(PC\left(\theta_{o}\right)\cos\left(\theta_{o}\right)\right)}^{2}$$
(17)$$b=2\sum_{o}\left(PC\left(\theta_{o}\right)\cos\left(\theta_{o}\right)\right)\left(PC\left(\theta_{o}\right)\sin\left(\theta_{o}\right)\right)$$
(18)$$c=\sum_{o}{\left(PC\left(\theta_{o}\right)\sin\left(\theta_{o}\right)\right)}^{2}$$

Then, the angle of the principal axis indicating the direction information of the feature, which is the axis corresponding to the minimum moment, ψ, is given by:(19)$$\psi =\frac{1}{2}\arctan\left(\frac{b}{a-c}\right)$$

The minimum and maximum moments, $m_{\psi }$ and $M_{\psi }$, respectively, are obtained by:(20)$$m_{\psi }=\frac{1}{2}\left(c+a-\sqrt{b^{2}+{\left(a-c\right)}^{2}}\right)$$
(21)$$M_{\psi }=\frac{1}{2}\left(c+a+\sqrt{b^{2}+{\left(a-c\right)}^{2}}\right)$$

$m_{\psi }$ and $M_{\psi }$ can be used for corner and edge feature detection, respectively. Therefore, local maxima detection and non-maximal suppression are performed to obtain corners on the minimum moment map, and features from accelerated segment test (FAST) is selected to detect edge feature points on $M_{\psi }$. Following this method, corner and edge features can be integrated for feature matching [51,52]. Considering the location error when generating the building change proposals, we conservatively expanded the proposals by morphological dilation operation, i.e. the rectangle element of 7 × 7 pixels, as the valid region. Figure 7 shows the results of feature detection in the valid region.

#### 2.2.2. Structural Feature Descriptor for Matching

Upon completion of the above feature points extraction, we needed to build a descriptor to represent and distinguish the feature points as suitably as possible. Classical feature descriptors generally use image intensity or gradient distribution to construct feature vectors. However, since both intensity and gradient are very sensitive to non-linear radiometric differences (NRD), these descriptors are not suitable for matching task. Intuitively, given the advantage of PC, using PC instead of intensity or gradient is suitable for feature description. Ye et al. [53] used the amplitude and orientation of phase congruency to build the HOPC descriptor inspired by the histogram of gradient (HOG). The HOPC descriptor captures the structural features of images and is more robust to illumination changes. Since structural properties are relatively independent of intensity distribution patterns in images, this descriptor can be used to match two images having significant NRD as long as they both have similar shapes. We first used the novel structural feature descriptor HOPC to perform matching, and the detailed description can be found in Ye et al. [53]. In terms of feature matching, the correspondence problem is simplified into interregional matching between related domains. For a query feature, the search scope is narrowed down to a buffer within 35 meters, which is included in the valid regions of search image. Within the two feature subsets, we took the normalized correlation coefficient (NCC) of the HOPC descriptors as the similarity metric $HOPC_{ncc}$ for roof matching, which is defined as [53]:(22)$$HOPC_{ncc}=\frac{\sum_{k=1}^{n}\left(V_{A}\left(k\right)-{\bar{V}}_{A}\right)\left(V_{B}\left(k\right)-{\bar{V}}_{B}\right)}{\sqrt{\sum_{k=1}^{n}{\left(V_{A}\left(k\right)-{\bar{V}}_{A}\right)}^{2}\sum_{k=1}^{n}{\left(V_{B}\left(k\right)-{\bar{V}}_{B}\right)}^{2}}}$$
where $V_{A}$ and $V_{B}$ are HOPC descriptors of the image regions A and region B, respectively, and ${\bar{V}}_{A}$ and ${\bar{V}}_{B}$ denote the means of $V_{A}$ and $V_{B}$, respectively.

After putative matches are found between the bi-temporal changed building proposals, the matches are combined with robust filtering strategies, including ratio test and random sample consensus (RANSAC), to finally produce geometrically-consistent matches. $HOPC_{ncc}$ successfully matches the bi-temporal aerial image pairs, as shown in Figure 8. In the matching processing, template windows of different sizes ($s_{w}\times s_{w}$) have an effect on the correct matching ratio, and the template window is constructed using blocks having an α degree of overlap. Each block consists of (m×m) cells containing n×n pixels, and each cell is divided into β orientation bins. Thus, $s_{w}$, α, m, n, and β are the parameters to be tuned. Their influences, which are set to 60, 1/2, 3, 4, and 8 in this study, respectively, were tested by Ye et al. [53].

### 2.3. Co-Refinement for Final Building Change Detection

Through the process of patch-based matching, the corresponding relationship of the roofs between the bi-temporal aerial images was determined. It is natural to think that the matching inconsistent unchanged buildings can be removed with the assistance of the bi-temporal changed building proposals. Meanwhile, considering when the separate and independent strategy is performed on the bi-temporal images, respectively, will result in duplicate work and the inadequate use of information. Therefore, in this paper, in order to eliminate mislabeled building changes in the bi-temporal aerial images simultaneously and provide high quality information, we employed a densely connected CRF model, integrating changed building proposals, matched rooftops, and appearance similarity of bi-temporal images for co-refinement. In this way, all building change information in the bi-temporal aerial images was gathered to form the detection result. Moreover, the spatial correspondence was inherently yielded between changed objects with the association of the bi-temporal changed building proposals, which can apparently be used to reveal the object-to-object changes for further type identification.

#### 2.3.1. Co-Refinement with CRF

A CRF is a form of a Markov Random Field (MRF) that directly defines the posterior probability, i.e., the probability of the output variables given the input data. The CRF is defined over the random variables $L=\left\{l_{1},l_{2},\cdots ,l_{p}\right\}$, where each $l_{p}\in \left\{0,1\right\}$, 0 for background and 1 for foreground, represents a binary label of the pixel p∈N={0,1,⋯,n} such that each random variable corresponds to a pixel. x denotes a joint configuration of these random variables, and I denotes the observed image data. Based on the general formulation in Krähenbühl et al. [54], a fully connected binary label CRF can be defined as:(23)$$E\left(x\right)=\underset{p\in N}{\sum }\psi_{p}\left(l_{p}\right)+\underset{p<q}{\sum }\psi_{pq}\left(l_{p},l_{q}\right)$$
where $l_{p}$ is the label taken by pixel p, N represents the set of all image pixels, and $\psi_{p}$ and $\psi_{pq}$ denote the unary and pairwise potentials, respectively.

The unary term $\psi_{p}\left(l_{p}\right)$ measures the cost of assigning a binary label $l_{p}$ to the pixel p. In this study, $\psi_{p}\left(l_{p}\right)$ is calculated for each pixel by the fusion of the appearance potential and the estimated matching between the bi-temporal changed building proposals:(24)$$\psi_{p}\left(l_{p}\right)=\{\begin{array}{c} -\log\left(\max\left(P\left(l_{p}^{t1}\right),P\left(l_{p}^{t2}\right)\right)\cdot M_{p}\right),\;if\;l_{p}=1\\ -\log\left(\max\left(P\left(l_{p}^{t1}\right),P\left(l_{p}^{t2}\right)\right)\cdot \left(1-M_{p}\right)\right),\;otherwise \end{array}$$
(25)$$P\left(l_{p}^{t}\right)=\frac{P\left(\Theta_{l_{p}}^{t},I_{p}^{t}\right)}{P\left(\Theta_{0}^{t},I_{p}^{t}\right)+P\left(\Theta_{1}^{t},I_{p}^{t}\right)}$$
where $P\left(l_{p}^{t}\right)$ is the probability of assigning a binary label $l_{p}$ to the pixel p in the tth period, which can be computed independently for each pixel by a classier that produces a distribution over the label assignment $x_{i}$ deduced from the initial SBL; $P\left(\Theta_{0}^{t},I_{p}^{t}\right),\;P\left(\Theta_{1}^{t},I_{p}^{t}\right)\in \left(0,\infty \right)$ represents the probability density value of a pixel color $I_{p}$ belonging to the background color model $\Theta_{0}$ and the foreground color model $\Theta_{1}$ at the tth period, respectively. We used Gaussian Mixture Models (GMMs) and followed the implementation details in Cheng et al. [55] to estimate the probability density values according to the initial SBL. We used a distance transform function to evaluate the unchanged portions in each connected region of the bi-temporal changed building proposals where the matched points reside and take this as $M_{p}$, which is defined as:(26)$$M_{p}=\{\begin{array}{c} d\left(p,q\right)/D_{max},\;if\;d\left(p,q\right)<D_{max}\;and\;p\in \Gamma_{q}\\ 0,\;else\;if\;CBP_{p}^{t1}=CBP_{p}^{t2}=1\;and\;p\in \Gamma_{q}\\ \max\left(CBP_{p}^{t1},CBP_{p}^{t2}\right),\;otherwise \end{array}$$
where d(p,q) is the distance of the arbitrary pixel p to the nearest matched pixel q, $D_{max}$ is the maximum distance (i.e. 5.0 m), $\Gamma_{q}$ denotes the connected region which contains the matched point q, and $CBP_{p}^{t1},CBP_{p}^{t2}$ represent the value of pixel p in the changed building proposals at time $t_{1}$ and $t_{2}$, respectively.

The pairwise term $\psi_{pq}\left(x_{p},x_{q}\right)$ encourages similar and nearby pixels to take consistent labels. We used a contrast-sensitive two kernel potential: (27)$$\psi_{pq}=g\left(p,q\right)\left[l_{p}\neq l_{q}\right]$$
(28)$$g\left(p,q\right)=w_{1}g_{1}\left(p,q\right)+w_{2}g_{2}\left(p,q\right)$$
where the Iverson bracket [·] is 1 for a true condition and 0 otherwise, $w_{1}\;\mathrm{and}\;w_{2}$ are the weight coefficients controlling the impacts of color and spatial configuration, which are defined in terms of color vectors $I_{p}^{t},\;I_{q}^{t}$ and position coordinates $c_{p},\;c_{q}$, respectively:(29)$$g_{1}\left(p,q\right)=\exp\left(-\frac{\max\left({||I_{p}^{t1}-I_{q}^{t1}||}^{2},{||I_{p}^{t2}-I_{q}^{t2}||}^{2}\right)}{\theta_{\alpha }^{2}}-\frac{{||c_{p}-c_{q}||}^{2}}{\theta_{\beta }^{2}}\right)$$
(30)$$g_{2}\left(p,q\right)=\exp\left(-\frac{{||c_{p}-c_{q}||}^{2}}{\theta_{\mu }^{2}}\right)$$
where $g_{1}\left(p,q\right)$ models the appearance similarity and encourages nearby pixels with similar color to have the same binary label and $g_{2}\left(p,q\right)$ encourages smoothness and helps remove small isolated regions. The degree of similarity, nearness, and smoothness are controlled by $\theta_{\alpha }$, $\theta_{\beta }$, and $\theta_{\mu }$, respectively. Intuitively, $\theta_{\beta }\gg \theta_{\mu }$ should be satisfied if the color configuration manages the long-range connections and the spatial configuration measures the local smoothness.

As shown in Figure 9, we integrated bi-temporal changed building proposals and the matched points (Figure 9a,b) to calculate $M_{p}$ (Figure 9c) for the unary term, and the converged appearance similarity (Figure 9d) was obtained for the pairwise term by picking up the maximal color contrast of the bi-temporal images. By gathering unary and pairwise terms, co-refinement is performed in a densely connected CRF for high quality building change detection. On the basis of all changed building proposals in the bi-temporal images (Figure 9e), we divided the images into foreground and background segments representing the final building changes and others, respectively (Figure 9f). 

The proposed fully connected model improves two aspects of the detection quality. First, the pairwise potential is defined over all pairs of pixels, which allows the model to capture long-range interactions; thus, the segmentation of objects associated with long-range context is augmented. Second, unlike the commonly used pairwise potential where only color contrast is considered, the proposed pairwise term incorporates both color contrast of bi-temporal images and spatial distance. Therefore, the proposed model is able to generate more accurate segmentation of objects with noise caused by sampling and proximity to other objects.

#### 2.3.2. Type Identification of Changed Buildings

After obtaining the bi-temporal changed buildings using the above processing, we used the spatial correspondence analysis to further classify them as “newly built”, “demolished”, and “changed”. We performed overlay analyses between the final detection result and changed building proposals at each period to delete the non-overlapping objects, which are generally false alarms, such as image noise, obstruction, and isolated pixels. We conducted a union operation on the two generated maps and assigned a change type to each object. Figure 10 shows the step of determining change type, and the rules are summarized in Table 1. 

## 3. Experimental Results

To verify the effectiveness and generality of the proposed algorithm, two datasets of bi-temporal aerial images were collected for experiments. In this work, bi-temporal datasets were georeferenced and registered. However, the relief displacements of the buildings usually vary considerably with location and cannot be eliminated by geo-rectification in the images produced. After a brief description of the datasets and accuracy assessment measurements, the experimental results of the proposed method are provided below.

### 3.1. Study Site and Dataset Description

Dataset 1 is located in the urban area of Chongqing, China, with a valid area of approximately 6.3 km^2^ (~6,315,533 m^2^). The aerial images (7464 × 7629 pixels) were acquired in 2012 and 2013, with three multispectral bands and a resolution of 0.5 m. An overview of the dataset and its enlarged subsets are shown in Figure 11. This area covered a typical urban environment with sparse housing, high-rise buildings, commercial districts, and industrial areas. The buildings are differently distributed and vary in size and structure, and a few buildings are surrounded by trees.

The second dataset, Dataset 2, is situated in Ningbo, Zhejiang Province, China, with 2756 × 2744 pixels. This area is a complex suburban scene including a residential area with scattered high-rise buildings, an industrial area with dense large buildings, and some small buildings densely aligned along the street, as shown in Figure 12. The aerial images include three visible bands (RGB) with 0.5 m ground sample distance (GSD), and were acquired in 2015 and 2017. 

### 3.2. Building Change Detection Results

In this study, after generating the changed building proposals, we implemented patch-based matching and the co-refinement with CRF to detect building changes. The building change truths were prepared in advance by manually interpreted delineation and then used to evaluate the accuracy. During the experiment, the parameters of the two datasets were the same as follows: the parameter of changed building proposal generation λ=1.0 and parameters for the fully connected CRF model are $\left(w_{1},w_{2},\theta_{\alpha },\theta_{\beta },\theta_{\mu }\right)=\left(6.0,10.0,32.0,20.0,3.0\right).$ Meanwhile, in order to keep the buildings in the local scale, we performed patch-based roof matching by dividing the full image into many blocks. In addition, considering that there may be some differences among the buildings caused by the random division, we further determined the corresponding relationship of the remaining roofs on the basis of the previous matching by overlapping sliding windows. Figure 13 and Figure 14 show the change truths of buildings and the detected results of the proposed method in Datasets 1 and 2, respectively. In these changes, red implies the “newly built”, green represents “demolished”, and green denotes “changed”. The detected results are essentially consistent with the ground truth, proving that the proposed method is effective.

To observe the results in much more detail, Figure 15 shows the change detection results of six enlarged sub-regions with different types of buildings and scenes. The detailed subareas in Figure 15A–D and Figure 15E,F correspond to Datasets 1 and 2, respectively. The two columns on the left show the bi-temporal aerial images, the third column depicts the results of the proposed method, and the last column provides the building change truths. 

Subset A is an industrial area mainly composed of low-rise buildings with uneven distribution that vary in size and structure, with some makeshift shanties for carpenters constructing new buildings. In this area, the bi-temporal buildings have low relief displacements, and their rooftops are homogeneous. Sub-region B is a residential area with three communities, where some temporary low-rise sheds disappear with the completion of some building projects. In each community, the buildings are uniformly distributed and are quite similar, with certain differences between them. In this subarea, although the bi-temporal buildings have different relief displacements for each community, their appearance and geometric displacements are consistent. In the two regions in sub-regions C and D, the buildings are also uniformly distributed and have a certain resemblance to the periphery. For subarea C, most buildings in the bi-temporal aerial images are high-rise with large relief displacements and their rooftops are homogeneous, but the geometric displacements are partly inconsistent due to the influence of image splicing at time $t_{2}$. There are various types of building changes as the buildings under construction are completed. Both low-rise buildings with minimal relief displacement and high-rise buildings with large relief displacements are present in subarea D. The rooftop of each building varies but the geometric displacements are relatively consistent with the surroundings. Subsets E and F include residential areas with scattered high-rise buildings and industrial districts with dense large buildings in a complex suburban scene, where buildings are mostly complex and different in size, structure, and distribution. In the two regions, both new added buildings and demolished buildings exist.

Notably, the chosen regions are representative scenes that are complex and highly challenging for building monitoring, and the results indicate that the presented approach is highly robust and suitable. Most changed buildings are successfully detected, and their positions are accurate. These findings were also verified by the following quality assessments.

### 3.3. Quality Assessments

In addition to visual illustration for the quality assessment of building change detection, we evaluated the object-level performance by counting the truly (*TD*), falsely (*FD*), and missing (*MD*) detections, and then calculated the correctness (*Corr*), completeness (*Comp*), and F-score (*F*_1_) measures [56], which are respectively defined as: (31)$$Corr=\frac{TD}{TD+FD}$$
(32)$$Comp=\frac{TD}{TD+MD}$$
(33)$$F_{1}=\frac{2Corr\cdot Comp}{Corr+Comp}$$

Under the prerequisite of change type being guaranteed to be consistent, we used an overlapping threshold of 70% to determine the number of *TD* and *MD,* as previously described in Ok et al. [57]. The overlap ratio was computed in terms of the number of pixels and a detected building was labeled as *TD* if at least 70% of the building overlapped with a reference building. Thus, *TD* represents true positives and corresponds to the number of changed buildings correctly classified in both ground truth and the detection result. *FD* represents false positives and corresponds to the total of buildings mislabeled as a change and changed buildings misclassified. *MD* represents false negatives and corresponds to the number of buildings mislabeled as no change. *F*_1_ measures the overall performance through the weighted harmonic of completeness and correctness. Figure 16 shows the corresponding results evaluation between the proposed method and ground truth in two datasets, in which *TD*, *FD* and *MD* are displayed in yellow, pink, and cyan, respectively. The detailed evaluation results are depicted in Table 2 and Table 3.

Table 2 and Table 3 show that the proposed method performs well in the two datasets, and is applicable for suburban areas and complex urban scenes including dense scattered building areas (i.e., villages in the city) and narrow streets. This result is mainly attributed to the semantic segmentation used in our method, which accurately extracts the building area, and the patch-based match effectively eliminates the inconsistent unchanged buildings. 

In Dataset 1, there were 299 correct detections, 24 missed detections, 16 detections misclassified by wrong change type, and 38 false detections. Correctness, completeness, and overall accuracy were 84.7%, 92.57%, and 88.46%, respectively. The missing changes were caused by imperfect change information (5/24), ineffective semantic segmentation for the buildings under construction (11/24), serious shelter of small buildings in the peripheral environment (1/24), and improper filtering on the shanties with a few pixels (7/24), as shown in Figure 17. The 16 misclassified building changes mainly resulted from the following aspects: interference of change occurring in the surrounding buildings and wrong semantic segmentation, as shown in Figure 18a,b. The main reason for buildings being wrongly labelled as changes is the errors in CD resulting from the renovated buildings and failed matches caused by the different relief displacements, as shown in Figure 18c,d.

Because of the complexity Dataset 2, a little decrease in the correctness and the overall performance occurred, but the completeness was still maintained. A total of 127 changes were detected correctly out of 138, resulting in a completeness of 92.03%. For 27 false detections, besides the above reasons, another factor was the interference of suspected building objects (i.e., regular surface, sunshade), as shown in Figure 19a. A few missed detections and misclassified detections occurred for Dataset 2. The main reasons for these were as follows: confused changed buildings and broken roofs composed of several small roofs, as shown in Figure 19b,c. 

## 4. Discussion

### 4.1. Parameter Selection

The proposed method mainly involves the following parameters: λ for proposal generation, and parameters for the fully connected CRF model. With Dataset 1 as an example, we fixed several other items and changed the item to be evaluated. The object-based statistics are shown in Figure 20.

λ in Equation (5) is a weight coefficient that measures the relative importance of the change information versus the image feature in the energy function of a segmentation based on graph cuts. The larger the value of λ, the greater the consideration of the image feature; and the smaller the λ, the more the focus on the change information in the process of proposal generation. The influence of λ on the *Corr*, *Comp*, and *F*_1_ in building change detection is shown in Figure 20a, and the experimental results reveal that λ has relatively little impact on the final result. This observation can be explained by SBL being introduced to impose a constraint on the energy function, and we set λ to 1.0 in this paper.

For the parameters used to calculate the pairwise potential in the fully connected CRF model, we initialized the parameters following the guidelines in Krähenbühl et al. [54] and then varied the parameters to search for the optimal settings on our own dataset. The experimental results revealed that $\theta_{\mu }$ has relatively little impact on the accuracy of the final result, as shown in Figure 20b, which was indicated in Krähenbühl et al. [54]. Thus, we set $\theta_{\mu }=3$, the same as suggested in Cheng et al. [55]. For $w_{1}$, $w_{2}$, $\theta_{\alpha }$, and $\theta_{\beta }$, the effects of parameter variation are shown in Figure 20c–f, respectively. $w_{1}$ and $w_{2}$ weight the impact of color and spatial configuration. Large values of $w_{1}$ lead to overemphasis on the color difference between adjacent pixels, which slightly improves the completeness, but at the cost of a drop in correctness, as shown in Figure 20c. Conversely, small values of $w_{2}$ neglect the spatial relationship between neighboring pixels, resulting in low correctness and high completeness, as shown in Figure 20d. $\theta_{\alpha }$ controls the color contrast of pairwise interaction. There is little change in accuracy when $\theta_{\alpha }\leq 32$, but when $\theta_{\alpha }$ is too high, some internal components that vary in color with the changed buildings would be retained, therefore producing more false positives and decreases in overall accuracy, as shown in Figure 20e. $\theta_{\beta }$ modulates the effects of spatial range. Accuracy increases as $\theta_{\beta }$ grows from 0 to 20, since the spatial smoothness helps to remove pixel-level noise in the local range of unchanged building. As a result, correctness increases substantially, as shown in Figure 20f. However, as $\theta_{\beta }$ continues to grow, accuracy tends to be stable.

### 4.2. Advantages and Disadvantages of the Proposed Algorithm

The results of this work are promising for building detection in challenging regions with complex scenes, indicating that the proposed method has several advantages. The proposed approach is suited for heterogeneous appearances due to complex structures, since it is associated with the same change information and takes full advantage of the extracted buildings of each phase on the basis of semantic segmentation in the process of generating the changed building proposals. As most building change detection approaches produce many inconsistent matches of unchanged buildings caused by the relief displacement and spectral variation, we employed patch-based feature matching with the structural HOPC descriptor to determine the corresponding roofs and identify the relevant buildings. Our proposed method can distinguish the type of change while quickly and accurately providing the change information on buildings.

Given its many benefits, our approach also has some limitations that must be minimized or overcome. In the determination of bi-temporal changed buildings, since the segmentation procedures depend on the change information and the building areas extracted by CNN-based semantic segmentation, some detections are missed when the changed building identifications are incomplete. During the process of co-refinement based on CRF, although the corresponding roofs representing the unchanged areas are obtained using an effective patch-based matching approach, matching the roofs lacking structure information is fairly difficult and may result in poor performance and even pseudo changes.

## 5. Conclusions

In this study, a novel patch-based matching approach was developed to perform building change detection with co-refinement based on CRF. The potential building changes at each time point are first obtained by a graph-cuts-based segmentation and integrating the change information and building areas using deep learning. These proposals are resistant to the heterogeneous appearances resulting from complex structures and reveal the spatially relevant changes. Then, patch-based feature matching with the structural HOPC descriptor is used to estimate the corresponding relation of the bi-temporal roofs in the interregional form. The advantage of interregional matching is that the number of reliable inlier correspondences increases because after shrinking the search range in scale space, more inlier correspondences are retained due to the distinctiveness relaxation, i.e., the constraint of feature distinctiveness is relaxed only inside the search scope. During the process of co-refinement based on CRF, the combination of the corresponding roofs and the information on potential building changes eliminates the matching of inconsistent unchanged buildings caused by the relief displacement and spectral variation. In addition, our method can distinguish the type of change while quickly and accurately providing the information on buildings that have changed. Extensive experiments showed that the proposed method is effective and robust for the change detection of buildings using aerial images with complex urban scenes. However, the proposed algorithm is sensitive to building extraction with semantic segmentation and CD technology. The missed detections caused by semantic segmentation and CD cannot be recovered in the subsequent optimization, and patch-based matching does not eliminate the false-positive buildings that are completely or seriously occluded by trees or shade due to the limited structure of the relevant buildings. 

In future studies, the following approaches could be considered: an assistant dense matching for whole roof localization, and using three-dimensional building change detection approaches, end-to-end BCD based on deep learning, and applying an algorithm for shadow elimination in an image enhancement step.

## Figures and Tables

**Figure 1 sensors-19-01557-f001:**
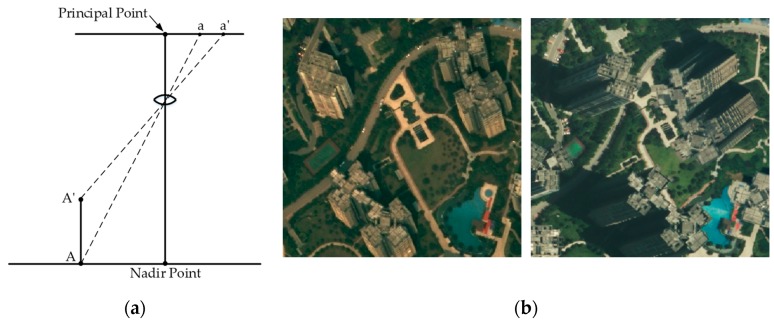
Underlying problems in building change detection: (**a**) Illustration of relief displacement and (**b**) image pairs including high-rise inclination buildings with greatly different relief displacements.

**Figure 2 sensors-19-01557-f002:**
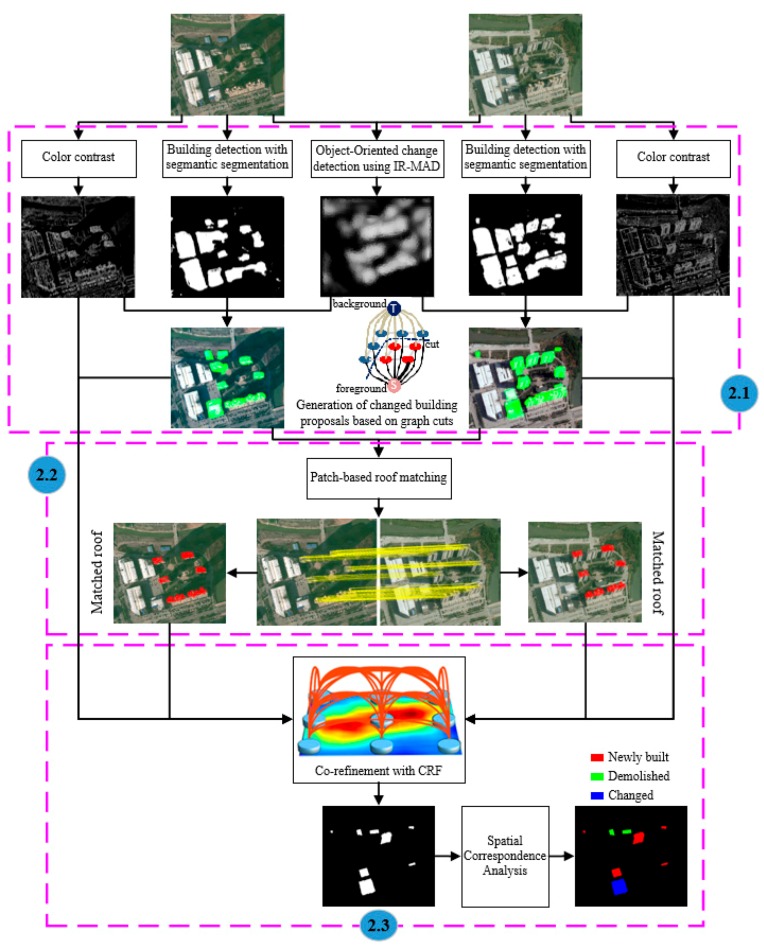
Flowchart of the proposed patch-based matching for the building change detection method.

**Figure 3 sensors-19-01557-f003:**
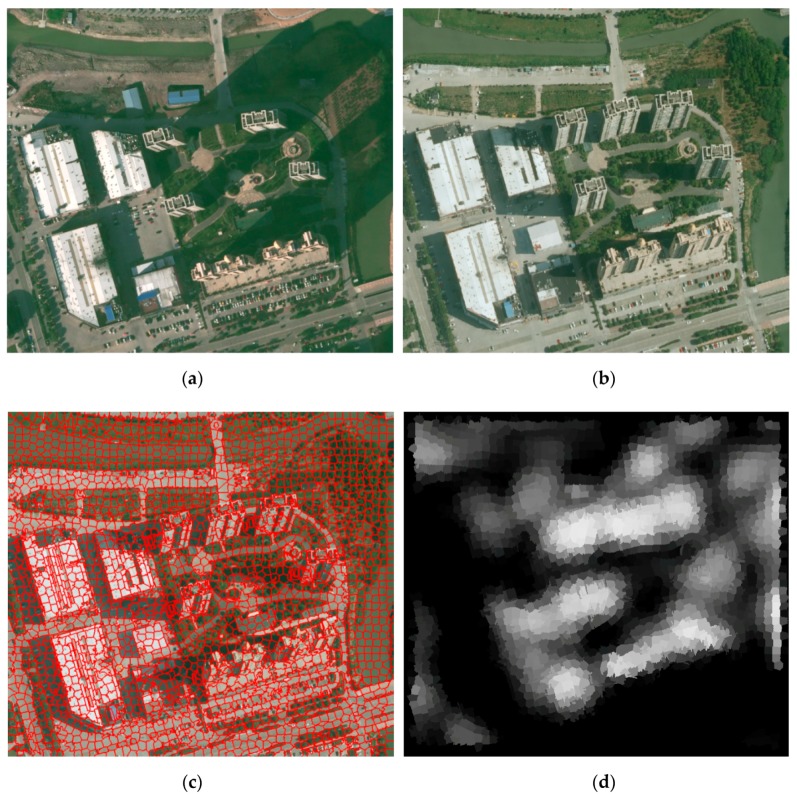
Bi-temporal images, the corresponding image-object, and change confidence map. (**a**) The RGB aerial image at time *t*_1_, (**b**) the RGB aerial image at time *t*_2_, (**c**) superposition of the bi-temporal image-object and the RGB aerial image at time *t*_2_, and (**d**) the change confidence map.

**Figure 4 sensors-19-01557-f004:**
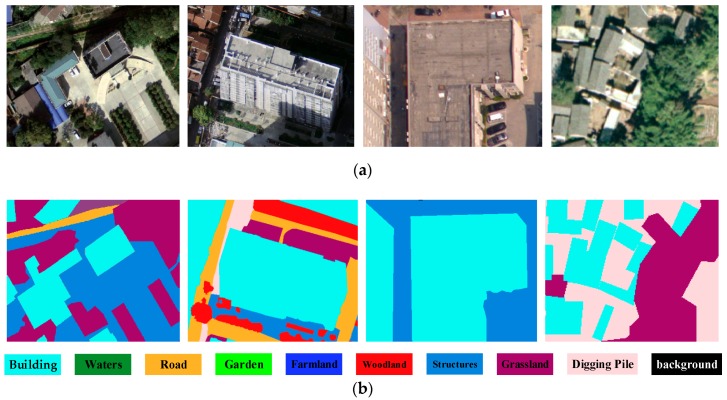
Examples of the training sample on the EvLab-SS dataset. (**a**) The RGB images, and (**b**) the corresponding label data.

**Figure 5 sensors-19-01557-f005:**
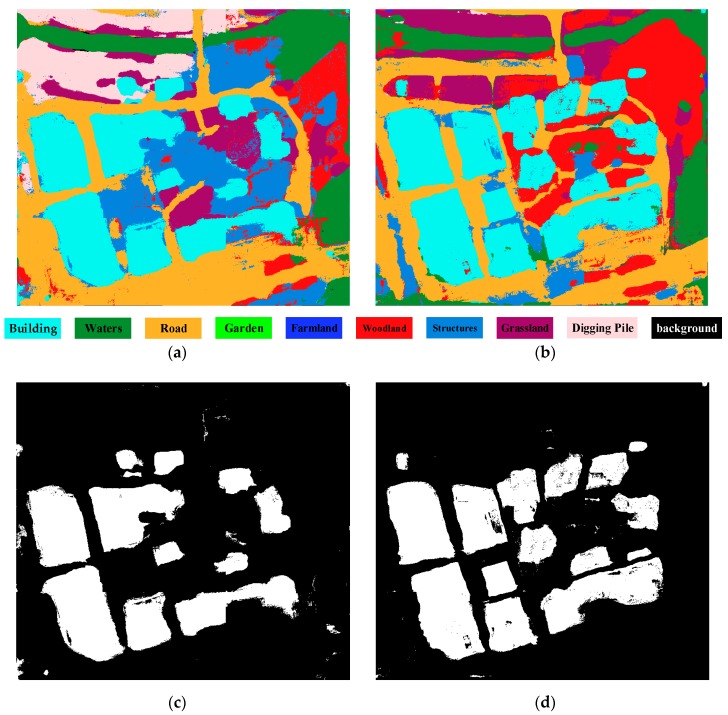
Results of sematic segmentation and building extraction. Sematic segmentation for the aerial image at (**a**) time *t*_1_ and (**b**) at time *t*_2_; sematic building label for the aerial image at (**c**) time *t*_1_ and (**d**) at time *t*_2._

**Figure 6 sensors-19-01557-f006:**
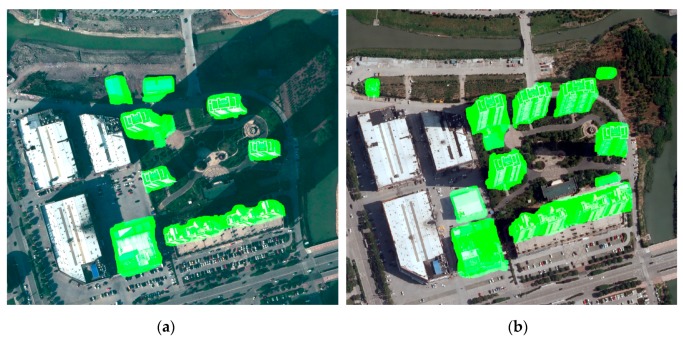
Changed building proposals in the bi-temporal aerial images at (**a**) time *t*_1_ and (**b**) at time *t*_2._

**Figure 7 sensors-19-01557-f007:**
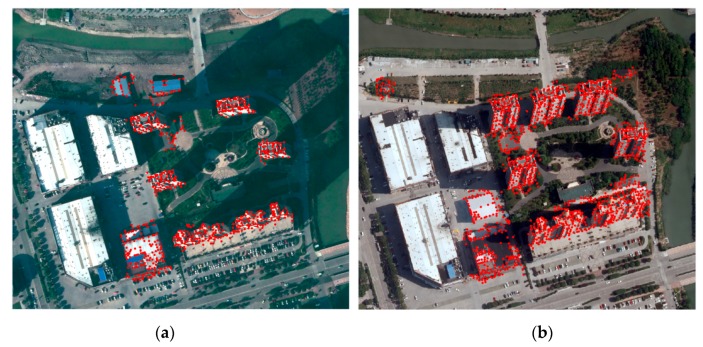
Feature detection. (**a**) Feature points detected in building change proposals at times *t*_1_ and (**b**) *t*_2_.

**Figure 8 sensors-19-01557-f008:**
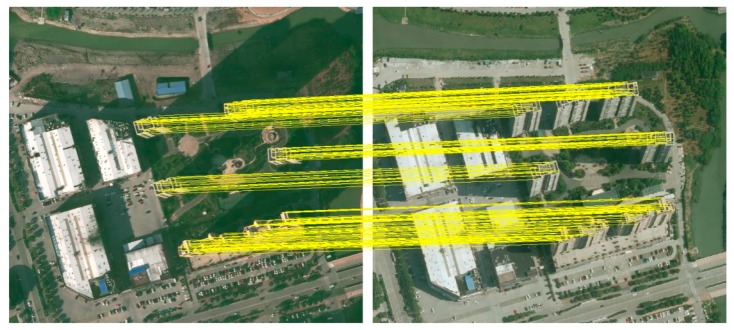
Results of patch-based matching using HOPC.

**Figure 9 sensors-19-01557-f009:**
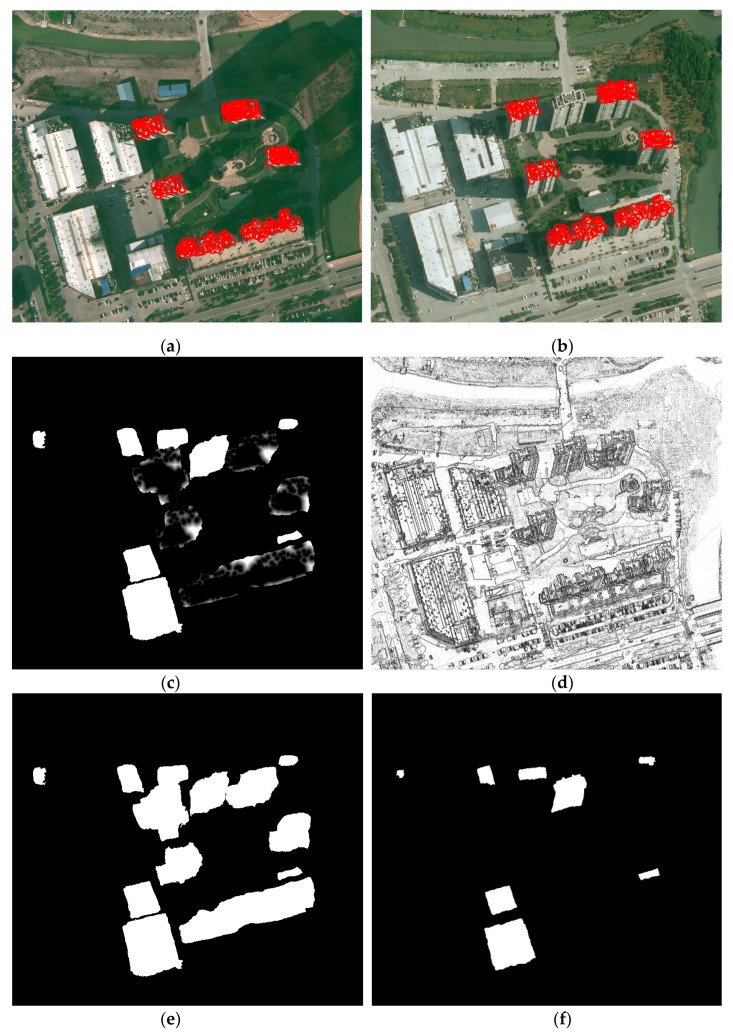
Co-refinement with CRF. (**a**) The buffers of the matched points on the roofs of images at times *t*_1_ and (**b**) *t*_2_; (**c**) the unchanged portions in each connected region $M_{p}$; (**d**) the converged appearance similarity of the bi-temporal images; (**e**) all changed building proposals in the bi-temporal images and (**f**) the final building change result.

**Figure 10 sensors-19-01557-f010:**
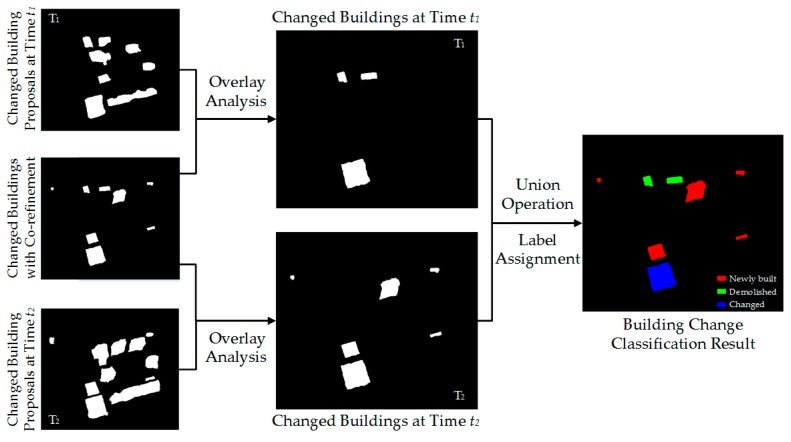
Step of determining the change type of a building using spatial correspondence analysis.

**Figure 11 sensors-19-01557-f011:**
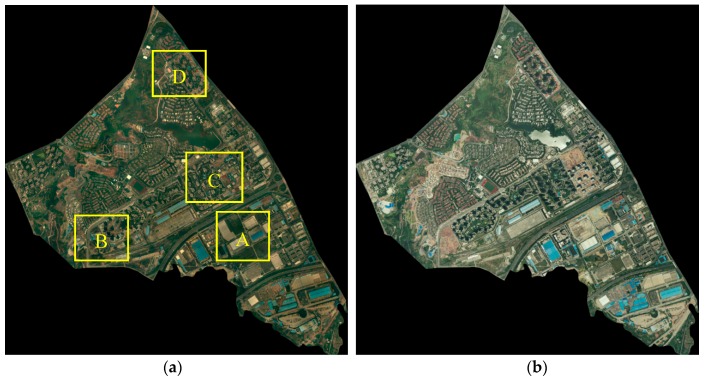
Overview of Dataset 1: the aerial image at (**a**) time *t*_1_ and (**b**) at time *t*_2_.

**Figure 12 sensors-19-01557-f012:**
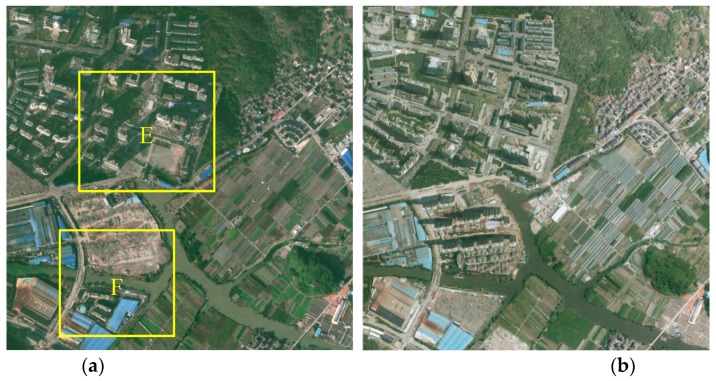
Overview of Dataset 2: aerial images at time (**a**) *t*_1_ and (**b**) *t*_2_.

**Figure 13 sensors-19-01557-f013:**
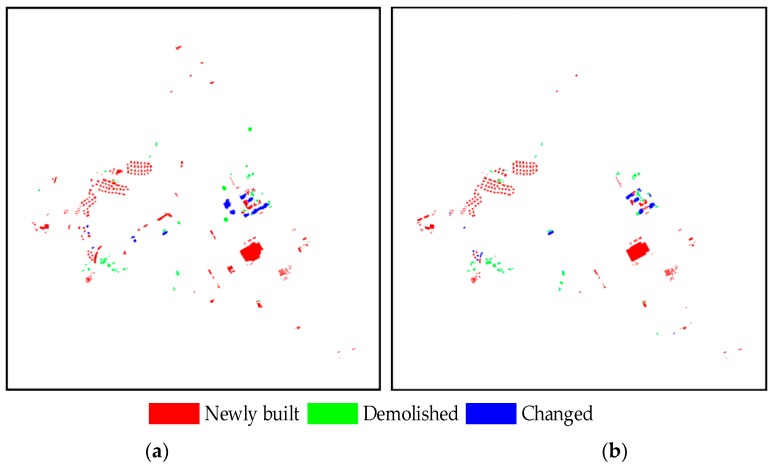
Results of building change detection using the (**a**) proposed method and (**b**) ground truth on Dataset 1.

**Figure 14 sensors-19-01557-f014:**
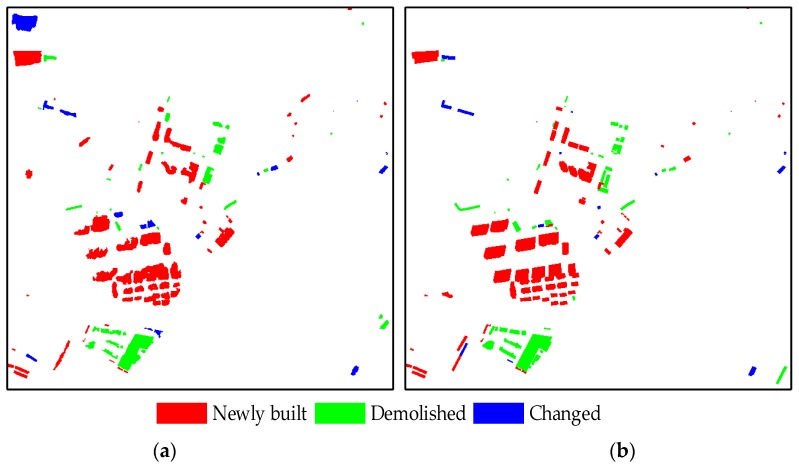
Results of building change detection using (**a**) the proposed method and (**b**) ground truth on Dataset 2.

**Figure 15 sensors-19-01557-f015:**
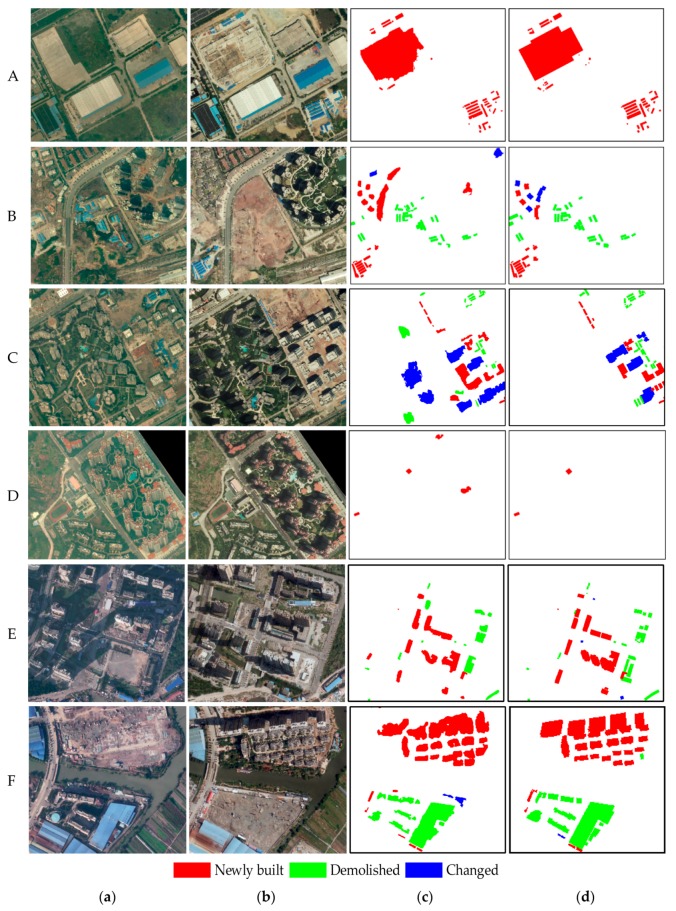
Amplified details of six typical regions: (**a**) Enlarged image subareas at time *t*_1_, (**b**) enlarged images at Time *t*_2_, (**c**) results of the proposed method, and (**d**) ground truth.

**Figure 16 sensors-19-01557-f016:**
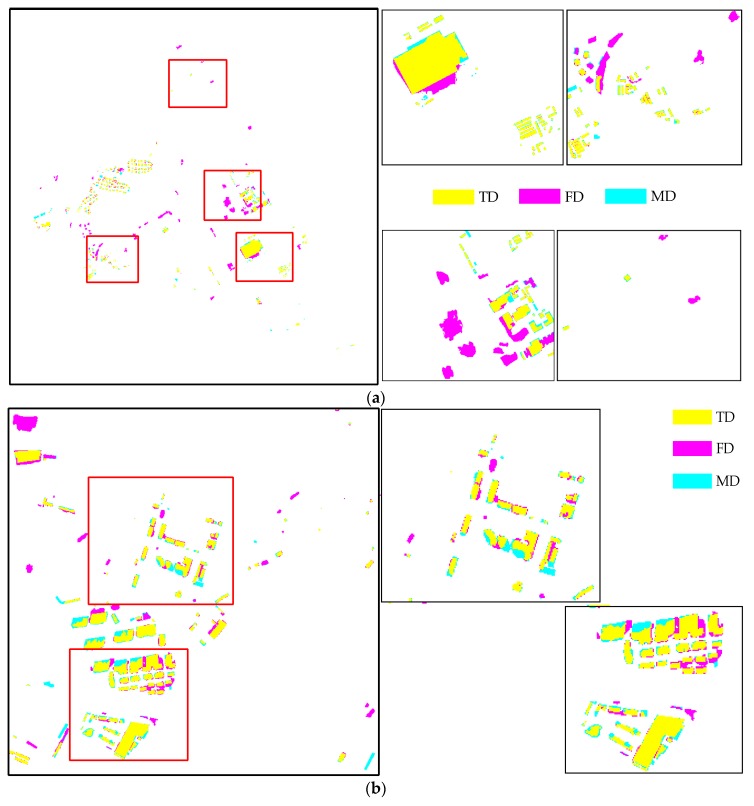
Corresponding results evaluations of the two datasets and enlarged subsets for (**a**) Dataset 1 and (**b**) Dataset 2.

**Figure 17 sensors-19-01557-f017:**
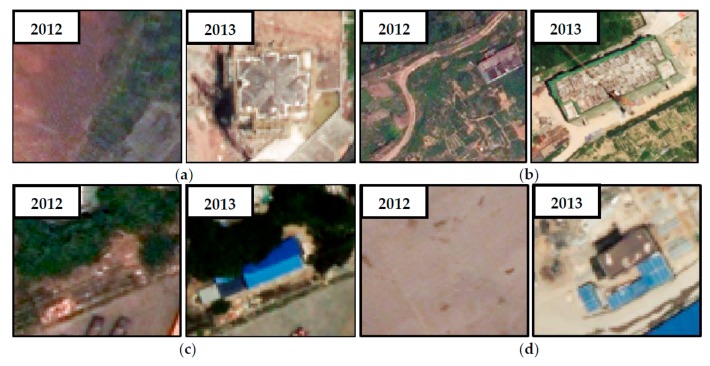
Examples of missing changes in Dataset 1: (**a**) missed detection caused by imperfect change information, (**b**) missed detection caused by ineffective semantic segmentation for the buildings under construction, (**c**) missed detection caused by serious sheltering of small buildings in the peripheral environment, and (**d**) missed detection caused by improper filtering of the shanties with a few pixels.

**Figure 18 sensors-19-01557-f018:**
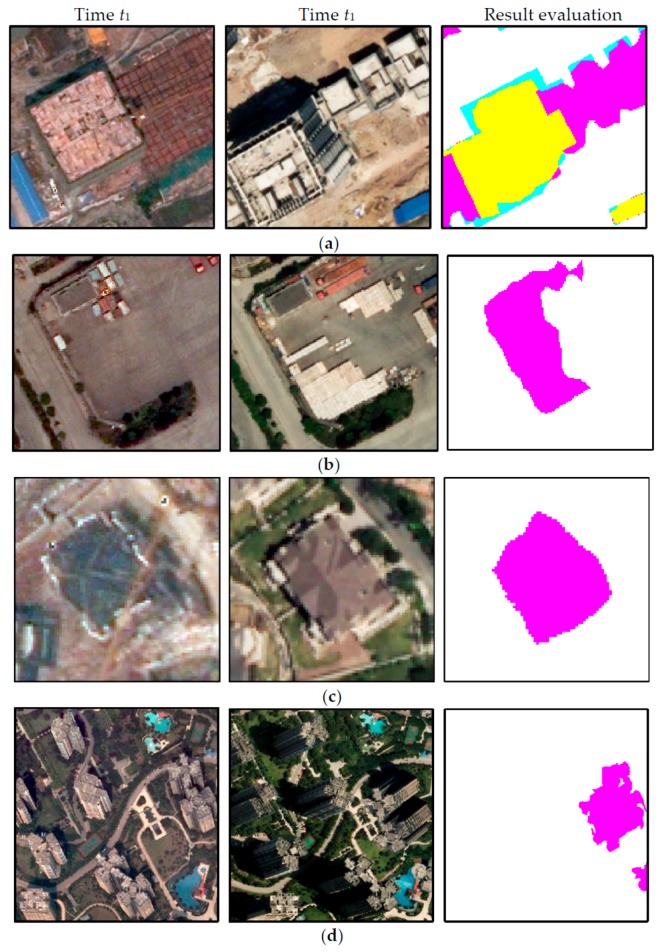
Examples of false detections in Dataset 1: (**a**) misclassified detection resulting from interference of change occurring in the surrounding buildings and (**b**) wrong semantic segmentation. (**c**) Wrong detection caused by the errors in CD resulting from the renovated buildings and (**d**) failed matching resulting from the different relief displacements.

**Figure 19 sensors-19-01557-f019:**
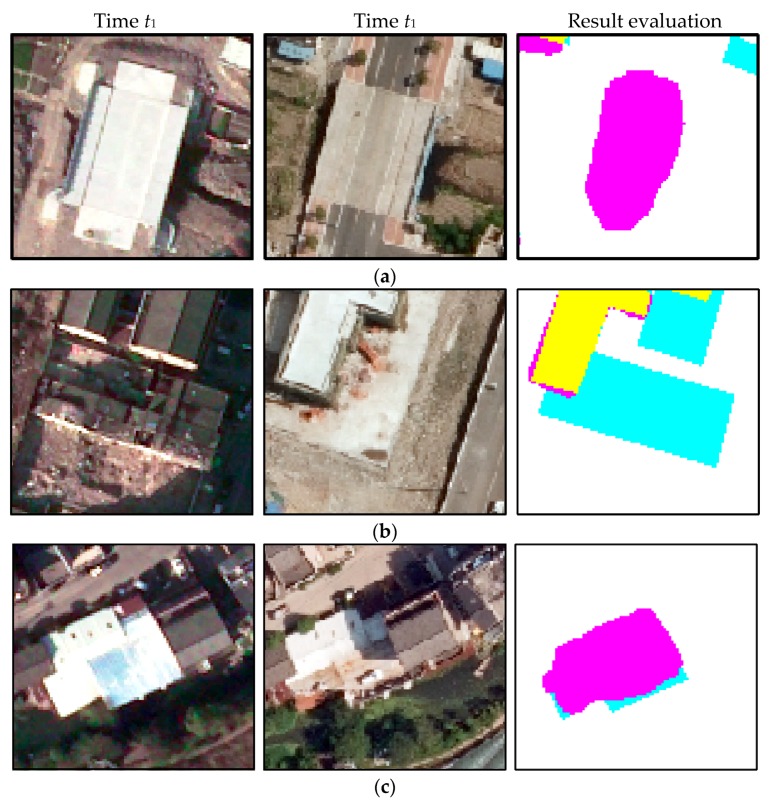
Limitations to the proposed method for Dataset 2: (**a**) wrong detection caused by the interference of suspected building objects, (**b**) missed detection caused by confused changed buildings, and (**c**) misclassified detection caused by broken roofs.

**Figure 20 sensors-19-01557-f020:**
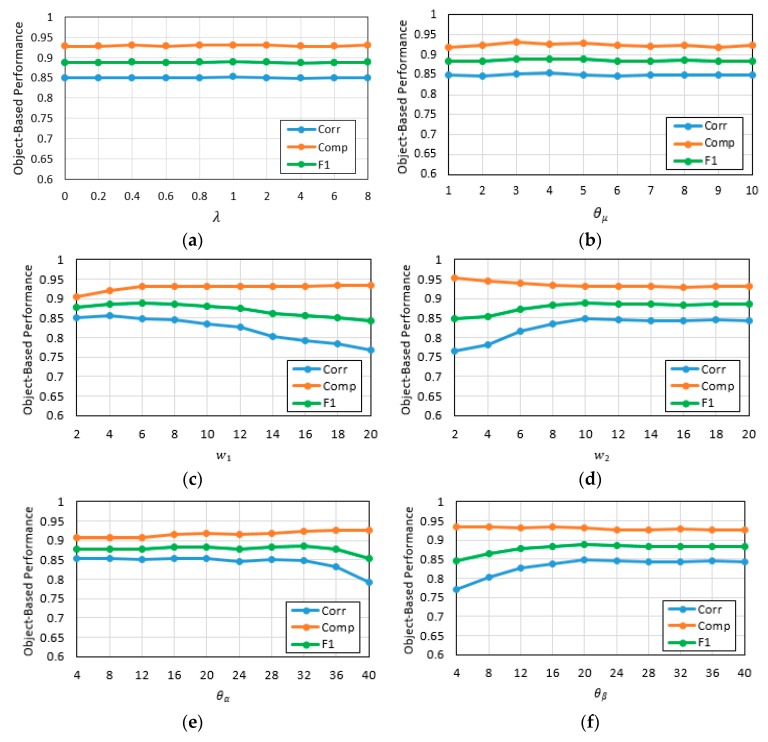
Object-based statistics of building change detection with different parameters. (**a**) Object-based statistics of building change detection with $\left(w_{1},w_{2},\theta_{\alpha },\theta_{\beta },\theta_{\mu }\right)=\left(6.0,10.0,32.0,20.0,3.0\right)$ and λ ranging from 0 to 8; (**b**) object-based statistics of building change detection with $\left(\lambda ,w_{1},w_{2},\theta_{\alpha },\theta_{\beta }\right)=\left(1.0,6.0,10.0,32.0,20.0\right)$ and $\theta_{\mu }$ ranging from 1 to 10; (**c**) object-based statistics of building change detection with $\left(\lambda ,w_{2},\theta_{\alpha },\theta_{\beta },\theta_{\mu }\right)=\left(1.0,10.0,32.0,20.0,3.0\right)$ and $w_{1}$ ranging from 2 to 20; (**d**) object-based statistics of building change detection with $\left(\lambda ,\;w_{1},\theta_{\alpha },\theta_{\beta },\theta_{\mu }\right)=\left(1.0,6.0,32.0,20.0,3.0\right)$ and $w_{2}$ ranging from 2 to 20; (**e**) object-based statistics of building change detection with $\left(\lambda ,w_{1},w_{2},\theta_{\beta },\theta_{\mu }\right)=\left(1.0,6.0,10.0,20.0,3.0\right)$ and $\theta_{\alpha }$ ranging from 4 to 40; (**f**) object-based statistics of building change detection with $\left(\lambda ,\;w_{1},w_{2},\theta_{\alpha },\theta_{\mu }\right)=\left(1.0,6.0,10.0,32.0,3.0\right)$ and $\theta_{\beta }$ ranging from 4 to 40.

**Table 1 sensors-19-01557-t001:** Change type determination with the guidance of a priori knowledge.

Time *t*_1_/Time *t*_2_	Changed Building	No Building Change
Changed Building	Changed	Demolished
No building change	Newly built	No building change

**Table 2 sensors-19-01557-t002:** Confusion matrix of the building change detection.

Analyzed Dataset	Proposed/Ground Truth	No. Building Change	Newly Built	Demolished	Changed
Dataset 1	No Building Change	0	13	9	2
	Newly Built	24	212	1	7
	Demolished	6	0	82	2
	Changed	8	4	2	5
Dataset 2	No Building Change	0	6	4	1
	Newly Built	8	71	0	3
	Demolished	5	0	47	2
	Changed	4	2	3	9

**Table 3 sensors-19-01557-t003:** The object-level performance of building change detection.

Analyzed Dataset	No. of Correctly Detected Buildings	No. of Wrongly Detected Buildings		*Corr* (%)	*Comp* (%)	*F*_1_ (%)
	*TD*	*FD*	*MD*			
Dataset 1	299	54	24	84.70	92.57	88.46
Dataset 2	127	27	11	82.46	92.03	86.98

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
