# Peer review of "Patch Matching and Dense CRF-Based Co-Refinement for Building Change Detection from Bi-Temporal Aerial Images"

_sensors, 2019, doi:10.3390/s19071557_

Round 1

Reviewer 1 Report

The paper presents a novel patch-based matching approach to perform building change detection with co-refinement based on CRF. The potential building changes at each time point are first obtained by a graph-cuts-based segmentation and integrating the change information and building areas using deep learning.

The research subject is an interesting one, dealing with the identification and monitoring of buildings from remotely sensed imagery which are considerably valuable for urbanization monitoring.

The paper has a good structure and organization, with a clear research methodology flowing in a logical manner and fully sustained by the presented results.

The paper requires a minor revision in order to be accepted for publication:

- at the page 14, lines 384-385, the sentence "An overview of the dataset and its enlarged subsets are shown in Figure 13" should be corrected - the right number of the figure is 9;

-at the page 14, lines 390-392, the sentence "This area is a complex suburban scene including a residential area with scattered high-rise buildings, an industrial area with dense large buildings, and some small buildings densely aligned along the street, as shown in Figure 7." should be corrected - the right number of the figure is 10;

at the page 16, lines 410-412, the paragraph "To observe the results in much more detail, Figure 12 shows the change detection results of six 411 enlarged sub-regions with different types of buildings and scenes. The detailed subareas in Figures 412 12A–12D and 12E,12F correspond to Datasets 1 and 2, respectively." should be corrected - the right number for the mentioned figure is 13;

- the equations numbering needs to be checked, because in section 3 the numbers are restarted from (2);

- after check of the eq. numbering, a text verification it is necessary;

- in section "4.1. Parameter Selection" the right number of the figure is 18. Also, in Figure 18 caption some explanations need to be added for figures 18 (d) to (e).

Author Response

Dear Reviewer:

First of all, we are sorry for bringing you some trouble because of our carelessness on labeling the figures. Thanks for your very insightful and valuable comments and recommendations that greatly help us improving the paper. In the revised copy, we made further modifications on some typos and grammar errors in the article. The detailed corrections are marked by using red text with the Track Changes mode in MS Word. If there are still any problems, please don’t hesitate to tell us. Thank you and best wishes!

The Main Corrections to the Paper:

1. We made careful modifications in the language of the article. The detailed corrections are marked with the Track Changes mode in MS Word.

2. The methodology is further elaborated and refined in Section 2.1 and 2.3.

3. All illustrations were labeled with correct figure/table numbers, captions and/or titles, and some were further improved.

Item-by-item responses

1. at the page 14, lines 384-385, the sentence "An overview of the dataset and its enlarged subsets are shown in Figure 13" should be corrected - the right number of the figure is 9;

Answer: Thanks for your comments and recommendations. It was corrected to the right number Figure 11 (the original version is Figure 9).

2. at the page 14, lines 390-392, the sentence "This area is a complex suburban scene including a residential area with scattered high-rise buildings, an industrial area with dense large buildings, and some small buildings densely aligned along the street, as shown in Figure 7." should be corrected - the right number of the figure is 10;

Answer: Thanks for bringing this to my attention. It was corrected to the right number Figure 12 (the original version is Figure 10).

3. at the page 16, lines 410-412, the paragraph "To observe the results in much more detail, Figure 12 shows the change detection results of six 411 enlarged sub-regions with different types of buildings and scenes. The detailed subareas in Figures 412 12A–12D and 12E,12F correspond to Datasets 1 and 2, respectively." should be corrected - the right number for the mentioned figure is 13;

Answer: Thanks for reminding me. They were corrected to the right number Figure 15 (the original version is Figure 13).

4. the equations numbering needs to be checked, because in section 3 the numbers are restarted from (2);

Answer: Thanks for your comments and recommendations. The equations numbering is corrected in Section 3

5. after check of the eq. numbering, a text verification it is necessary;

Answer: Thanks for your reminding. We have made a text verification after check of the eq. numbering.

6. in section "4.1. Parameter Selection" the right number of the figure is 18. Also, in Figure 18 caption some explanations need to be added for figures 18 (d) to (e).

Answer: Thanks for your comments and recommendations.

They have been corrected to the right number Figure 20 (the original version is Figure 18). “The small values of  neglect the spatial relationship between neighboring pixels, resulting in low correctness and high completeness, as shown in Figure 20d. controls the color contrast of pairwise interaction. There is little change in accuracy when , but when  is too high, some internal components that vary in color with the changed buildings would be retained, therefore producing more false positives and decreases in overall accuracy, as shown in Figure 20e.”

Reviewer 2 Report

This article presents a building change detection approach for bi-temporal aerial images using patch matching and dense CRF-based co-refinement of buildings. The topic is interesting and important in the remote sensing application fields especially for urban development monitoring. Motivation and experiment constructions are convincing. However, the methodology is a little bit complicated and hard to follow. Following comments might help improve the quality of the paper.

-       As one can see from Figure 2, the whole process of the methodology is quite complicated. I agree that all the steps are inevitable and necessary to carry out the building change detection, but it’s difficult to clearly understand the whole process at a glance. I would recommend including some figures regarding results after applying each sub-step with explanation.

-       P7L1: Please define DMSMR when it first appears in the manuscript. Moreover, please include a figure that is the result of building extraction with sematic segmentation (i.e., SBL), which is the next step of Figure 4.

-       Figure 4: Please mention the label of the colors in the figure.

-       P7, “training deep CNNs”: What is the training data to train the deep CNN for DMSMR? Please show some examples of the training data for building extraction. And please include information how many training data and labels (i.e., classes) are used for the training.

-       P9L245: RANSAC is used for estimating a geometric transform to get match points on buildings. However, this geometric transform is not appropriate for removing outliers. Aerial images (usually acquired by frame camera) have different magnitudes and directions of relief displacement according to the location of the building in image and the height of the building. Therefore, the matched points on changed buildings cannot be explained by just one geometric transform. It means that only buildings showing similar heights and located similar location of the images can be applied for the co-refinement. More diverse cases should be considered to improve the robustness of the proposed approach by modifying somehow.

-       P11, 2.3.1. Co-Refinement with CRF: This section is difficult to understand. What is the meaning of the co-refinement and what is the objective of the process? Some figures after carrying out the co-refinement might help understand the concept of the co-refinement.

-       Figure 8: it needs more explanation in detail of each sub-figures. For example, what is the left sub-image, and four sub-images in the middle?

-       Is the proposed approach applied for the full scene (i.e., Figure 9 and Figure 10) at once or for the subscene (i.e., from A to F in Figure 9 and 10) respectively? If the approach is applied to the full scene at once, again, only similar height buildings are able to be applied to change detection (i.e., errors like Figure 16d will occur severely).

-       P22: Some figure numbers are incorrect. Please check whether all figures are correctly numbered and explained in the text.

-       There are some typos and grammar errors. Please double check the entire manuscript carefully.

Author Response

Dear Reviewer:

First of all, we are sorry for bringing you some trouble because of the less detailed description on methodology and our carelessness on labeling the figures. Thanks for your very insightful and valuable comments and recommendations that greatly help us improving the paper. In the revised copy, we made further modifications on some typos and grammar errors in the article. In addition, the methodology is further elaborated and refined. The detailed corrections are marked by using red text with the Track Changes mode in MS Word. If there are still any problems, please don’t hesitate to tell us. Thank you and best wishes!

The Main Corrections to the Paper:

1. We made careful modifications in the language of the article by a native English speaker. The detailed corrections are marked with the Track Changes mode in MS Word.

2. The methodology is further elaborated and refined in Section 2.1 and 2.3.

3. All illustrations were labeled with correct figure/table numbers, captions and/or titles, and some were further improved.

Item-by-item responses

1. As one can see from Figure 2, the whole process of the methodology is quite complicated. I agree that all the steps are inevitable and necessary to carry out the building change detection, but it’s difficult to clearly understand the whole process at a glance. I would recommend including some figures regarding results after applying each sub-step with explanation.

Answer: Thank you for the good and insightful recommendation helping us make the structure of the article clear.

We have done the modification on Figure 2, and some figures regarding results were included after applying each sub-step with explanation, as shown in Section 2.

2. P7L1: Please define DMSMR when it first appears in the manuscript. Moreover, please include a figure that is the result of building extraction with sematic segmentation (i.e., SBL), which is the next step of Figure 4.

Answer: Thanks for bringing this to my attention.

  DMSMR represents the dual multi-scale manifold ranking, which have been defined at the first appearance in Section 2.1.2.

  We added the result of building extraction in Figure 5 (the original version is Figure 4).

3. Figure 4: Please mention the label of the colors in the figure.

Answer: Thanks for reminding me. We gave the class corresponding to the label of the colors in Figure 5 (the original version is Figure 4).

4. P7, “training deep CNNs”: What is the training data to train the deep CNN for DMSMR? Please show some examples of the training data for building extraction. And please include information how many training data and labels (i.e., classes) are used for the training.

Answer: Thanks for your comments and recommendations.

In the work, the EvLab-SS dataset from our team (It has been released in our website), which is designed for the high resolution pixel-wise classification task on real engineered scenes in remote sensing area, is used to train the deep CNN for DMSMR. The dataset is originally obtained from the Chinese Geographic Condition Survey and Mapping Project, and each image is fully annotated by the Geographic Conditions Survey standards. The average resolution of the dataset is approximately 4500 × 4500 pixels. The EvLab-SS dataset contains 11 major classes, namely, background, farmland, garden, woodland, grassland, building, road, structures, digging pile, desert and waters, and currently includes 60 frames of images captured by different platforms and sensors. We produce the training dataset by applying the sliding window with a stride of 128 pixels to the training images, thereby resulting in 48,622 patches with a resolution of 640 × 480 pixels. Similar methods are utilized on validation images, thus generating 13,539 patches for validation. In the paper, Figure 4 gives some examples of the training data for building extraction, as shown in the following. It was added in Section 2.1.2.

5. P9L245: RANSAC is used for estimating a geometric transform to get match points on buildings. However, this geometric transform is not appropriate for removing outliers. Aerial images (usually acquired by frame camera) have different magnitudes and directions of relief displacement according to the location of the building in image and the height of the building. Therefore, the matched points on changed buildings cannot be explained by just one geometric transform. It means that only buildings showing similar heights and located similar location of the images can be applied for the co-refinement. More diverse cases should be considered to improve the robustness of the proposed approach by modifying somehow.

Answer: Thanks for your comments and recommendations.

Yes, you are right. It is not appropriate for the full image to used RANSAC to estimate a geometric transform, because aerial images have different magnitudes and directions of relief displacement according to the location and height of the building in image. However, as pointed out in Section 2.2, since the buildings are quite similar on a local scale in terms of height, contour shape, and structure, we used filtering strategies for estimating geometric transforms and improving the stability of the process. Thus, the method of dividing into blocks is adopted in practice, which can make the buildings in the current range similar to some extent, and it is effective for general cases.

We divide the image into many blocks with a resolution of 1000 * 1000 pixels. Because some unchanged buildings with small relief displacements and non-building regions have been removed after the process of changed building proposals generation, we perform patch-based roof matching in the changed building proposals. It means that many interfering factors have been eliminated, most of the changed building proposals are similar in each block. Moreover, the search scope is narrowed down to a buffer within 35 meters for a query feature, and the matches are combined with ratio test and RANSAC to produce the corresponding points. Thus, it is feasible for buildings in the block to use RANSAC for getting match points and they can be applied for the co-refinement.

In addition, considering that there may be also some differences because of the random distribution, we can further determine the corresponding relationship of the roofs on the basic of the previous treatment by overlapping sliding window. Since some corresponding buildings have been eliminated by the matching in previous blocks, the similarity of the remaining buildings is high and the method is robust.

Although the approach is suitable to a certain extent, there will be a little errors. In the highly complex scenes, there are some unexpected cases, more diverse cases should be considered to further improve the robustness of the proposed approach. Thus, for future studies, we will try our best to solve the problem with non-rigid matching.

6. P11, 2.3.1. Co-Refinement with CRF: This section is difficult to understand. What is the meaning of the co-refinement and what is the objective of the process? Some figures after carrying out the co-refinement might help understand the concept of the co-refinement.

Answer: Thank you for the good and insightful comments helping us make the article technically correct and precise.

The meaning of the co-refinement is that all building change information from the bi-temporal aerial images is integrated to form the detection result, so that the spatial correspondence can be inherently yielded between changed objects with the association of the bi-temporal changed building proposals. In this way, the object-to-object changes can be revealed for further type identification.

The objective of the co-refinement is to eliminate mislabeled building changes in the bi-temporal aerial images simultaneously for avoiding numerous repeated calculations and provide high quality information. Through the process of patch-based matching, the corresponding relationship of the roofs between the bi-temporal aerial images was determined. It is natural to think that the matching inconsistent unchanged buildings can be removed with the assistance of the bi-temporal changed building proposals. Meanwhile, considering when the separate and independent strategy is performed on the bi-temporal images, respectively, will result in duplicate work and the inadequate use of information. Therefore, in the paper, we employed a densely connected CRF model integrating changed building proposals, matched rooftops, and appearance similarity of bi-temporal images for co-refinement.

The objective and meaning have been added in Section 2.3.

In order to help understand the concept of the co-refinement, some figures in this process and the co-refinement result are shown, and the related statements were added in Section 2.3.1. “As shown in Figure 9, we integrated bi-temporal changed building proposals and the matched points (Figures 9a and 9b) to calculate  (Figure 9c) for the unary term, and the converged appearance similarity (Figure 9d) was obtained for the pairwise term by picking up the maximal color contrast of the bi-temporal images. By gathering unary and pairwise terms, co-refinement is performed in a densely connected CRF for high quality building change detection. On the basis of all changed building proposals in the bi-temporal images (Figure 9e), we divided the images into foreground and background segments representing the final building changes and others, respectively (Figure 9f).”

7. Figure 8: it needs more explanation in detail of each sub-figures. For example, what is the left sub-image, and four sub-images in the middle?

Answer: Thanks for your recommendations.

  We have made a refinement on Figure 10 (the original version is Figure 8), and the corresponding explanation in detail of each sub-figures have been added in this Figure in Section 2.3.2.

8. Is the proposed approach applied for the full scene (i.e., Figure 9 and Figure 10) at once or for the subscene (i.e., from A to F in Figure 9 and 10) respectively? If the approach is applied to the full scene at once, again, only similar height buildings are able to be applied to change detection (i.e., errors like Figure 16d will occur severely).

Answer: Thanks you for the comments.

In practice, we divide the image into many blocks, most of the changed building proposals are regarded to be similar in each block. Meanwhile, considering that there may be also some differences because of the random distribution, we further determine the corresponding relationship of the roofs on the basic of the previous treatment by the overlapping way, as answered in 5.

We added an instruction about this processing way in Section 3.2. “Meanwhile, in order to keep the buildings in the local scale, we performed patch-based roof matching by dividing the full image into many blocks. In addition, considering that there may be some differences among the buildings caused by the random division, we further determined the corresponding relationship of the remaining roofs on the basis of the previous matching by overlapping sliding windows.”

9. P22: Some figure numbers are incorrect. Please check whether all figures are correctly numbered and explained in the text.

Answer: Thanks for bringing this to my attention.

All illustrations were labeled with correct figure/table numbers, captions and/or titles, and some were further improved.

10. There are some typos and grammar errors. Please double check the entire manuscript carefully.

Answer: Thanks for your comments and recommendations.

We made careful modifications in the language of the article. The detailed corrections are marked with the Track Changes mode in MS Word.

Round 2

Reviewer 2 Report

Most issues that I had have been properly addressed in the revised version of the manuscript.